# Characterization of Arbuscular Mycorrhizal Effector Proteins

**DOI:** 10.3390/ijms24119125

**Published:** 2023-05-23

**Authors:** María V. Aparicio Chacón, Judith Van Dingenen, Sofie Goormachtig

**Affiliations:** 1Department of Plant Biotechnology and Bioinformatics, Ghent University, 9052 Ghent, Belgium; maria.apariciochacon@psb.vib-ugent.be; 2Center for Plant Systems Biology, VIB, 9052 Ghent, Belgium

**Keywords:** secretome, effector proteins, arbuscular mycorrhizal fungi, interactomics, functional validation

## Abstract

Plants are colonized by various fungi with both pathogenic and beneficial lifestyles. One type of colonization strategy is through the secretion of effector proteins that alter the plant’s physiology to accommodate the fungus. The oldest plant symbionts, the arbuscular mycorrhizal fungi (AMF), may exploit effectors to their benefit. Genome analysis coupled with transcriptomic studies in different AMFs has intensified research on the effector function, evolution, and diversification of AMF. However, of the current 338 predicted effector proteins from the AM fungus *Rhizophagus irregularis*, only five have been characterized, of which merely two have been studied in detail to understand which plant proteins they associate with to affect the host physiology. Here, we review the most recent findings in AMF effector research and discuss the techniques used for the functional characterization of effector proteins, from their in silico prediction to their mode of action, with an emphasis on high-throughput approaches for the identification of plant targets of the effectors through which they manipulate their hosts.

## 1. Introduction

Approximately 80% of land plants associate with AMF [1,2]. Throughout this intimate root association, a complex transcriptional and physiological reprogramming of the plant is established to ensure the formation of arbuscules within the cortical cells that act as the functional unit of the symbiosis [3]. In these arbusculated cells, extensive trafficking of molecules takes place, with host plants delivering up to 20% of their fixed carbon to the symbiont in exchange for inorganic phosphate, nitrogen, water, and micronutrients [4,5,6,7].

Despite being the oldest plant symbiosis in the world [8], little is known about the fungal molecules that fine-tune this beneficial relationship in parallel to or downstream of the host perception of chitin oligosaccharides and lipochitooligosaccharides [9], also called Myc factors, an event governing fungal colonization [3]. One way microorganisms communicate with host cells is through the secretion and translocation of effector proteins, which have been intensively studied in plant-pathogen interactions [10,11,12] and are expected to modulate important aspects of the AM symbiosis as well [13,14]. These effector proteins have been shown to act in the plant cell apoplast or intracellularly, where they often interact with diverse host biomolecules to suppress immunity and allow fungal accommodation [10,15,16].

Several strategies, combining both in silico prediction and wet-lab experiments, have been proposed to highlight the effectors of interest for a particular plant-microbe interaction [17]. In the last decade, genomic and transcriptomic studies in different AMF strains, such as *Rhizophagus irregularis* [18,19,20,21,22], *Gigaspora rosea* [23,24], and *R. clarus* [21,25] have been published, underlining the existence of hundreds of potential effector genes, some of which are predicted to be conserved among different AMF strains [21,23]. Additionally, transcriptome analysis of different hosts and tissues colonized by *R. irregularis* shed light on the common and host-specific effectors used by this mycorrhizal fungus [26]. From these hundreds of potential effectors, a total of five *R. irregularis* effector proteins have been in-depth characterized for their role in AM symbiosis in the model legume *Medicago truncatula* [27,28,29,30,31]. Thus, hundreds of potential AMF effector proteins remain to be characterized that may play a relevant role in modulating the association.

Here, we review the most recent results in AMF effector research and discuss relevant techniques applied to the functional characterization of effector proteins, from bioinformatic predictions to their mode of action, focusing on high-throughput methods for the identification of plant targets through which the effectors control symbiosis.

## 2. Fungal Effector Protein Functions: Suppression of Defense and Niche Occupation

In general, pathogenic and beneficial plant-colonizing organisms utilize effector proteins to suppress immunity and to allow niche establishment [32,33]. To successfully colonize their plant hosts, microbes, including beneficial ones, must overcome the two major layers of plant immunity, which include membrane-localized and intracellular surveillance systems [17]. Cell surface perception of molecules derived from microbes or damaged plant cells via host pattern recognition receptors (PRRs) activates the pattern-triggered immunity (PTI) and downstream defense responses [34,35]. Depending on the nature of these molecules, they can be classified as microbe-associated molecular patterns (MAMPs), including chitin-containing molecules derived from the fungal wall, or as plant damage-associated molecular patterns (DAMPs) [36,37]. When microbes successfully overcome the first plant defense barrier, a second layer of intracellular immunity is initiated [38,39].

Microbes have developed multiple strategies to interfere with or avoid PTI recognition, including the secretion of specialized effector proteins [38]. Proteinaceous effectors are small secreted proteins (SSPs) that can regulate various aspects of the host physiology through selective binding to host plant macromolecules, such as proteins, DNA, and RNA [40]. A large collection of effectors involved in plant host immunity avoidance or suppression has been described for a variety of fungal and oomycete plant pathogens, such as the R×LR effector family or the Crinkler (CRN) effector in the pathogenic oomycete *Phytophthora infestans* [33,41,42,43,44]. In addition, effector proteins manipulate plant cells to establish the growth niche, and some change the nutrient status of the plant in favor of the pathogen, for instance by increasing the sugar efflux to boost microbial growth [45]. One of the best-known examples is the bacterial pathogen *Xanthomonas oryzae*, which infects *Oryza sativa* (rice) and secretes the PthXo1 effector into plant cells, where it activates the transcription of *SUGARS WILL EVENTUALLY BE EXPORTED TRANSPORTER 11* (*OsSWEET11*), which encodes a protein that exports sugars to the apoplast to feed the pathogen [45,46]. Although the direct association of fungal and oomycetes effector proteins to SWEET elements is lacking, the involvement of SWEET genes in fungal susceptibility has been recently elucidated [47,48,49]. For instance, the *Arabidopsis thaliana Atsweet14* mutants display low susceptibility to the infection by the necrotrophic fungus *Botrytis cinerea*, suggesting that this transporter is involved in feeding the fungus to support its development [47].

Based on structural analyses and functional validations, a wide range of in silico tools are now available to effectively predict fungal effector protein features. Key effector characteristics include (i) the presence of an N-terminal signal peptide of 18–30 amino acids that guides their conventional secretion outside the fungus [41]; (ii) the generally small size of less than 300 amino acids [50]; and (iii) the absence of a transmembrane domain to guarantee their presence in the extracellular space or inside the host cell [17,51]. To be directed to specific plant subcellular compartments, intracellular effectors often display specific amino acid sequences, such as nuclear localization signals (NLSs) or mitochondrial and chloroplast transient peptides [21,52]. Moreover, fungal effector proteins are frequently hallmarked by the presence of intrinsic disorder regions (IDRs) and the absence of characterized functional protein domains [53]. This lack of similarity to functionally characterized protein domains further challenges the identification of fungal effectors, because little information about their function can be extrapolated from the sequence [54].

Subsequently, the elucidation and characterization of the plant target are essential to explore the molecular processes in which effector proteins are involved. Indeed, studies of the targeted host proteins, DNA, or RNA regions of diverse fungal effectors have elucidated their mode of action in different plant hosts [42,55,56]. For example, the rust fungus *Melampsora larici-populina* secretes the nuclear-localized effector protein Mlp124478 that binds to the promoter region of the basic leucine zipper motif transcription factor (TF) TGA1a, boosting its transcription to suppress defense genes, such as the signaling gene *JAZ1* or the defense-related TF *WRKY18* in *Arabidopsis* [55]. Similarly, the two nuclear-localized effector proteins MoHTR1 and MoHTR2, from the rice blast pathogen *Magnaporte oryzae,* carry zinc finger DNA-binding domains that associate with the promoters of immunity-related genes encoding the plant TFs OsMYB4 and OsWRKY45 to repress their expression [42]. Another example of a functionally characterized effector is the arginine-rich RNA-binding effector protein Pst_A23 from the pathogenic fungus *Puccinia striiformis* of *Triticum aestivum* (wheat) [56]. Pst_A23 accumulates in nuclear speckles, where it manipulates the splicing of the leucine-rich repeat (LRR) receptor-like serine/threonine protein kinase mRNA *TaXa21-H* and the WRKY TF *TaWRKY53* by binding to the specific motifs M1 (5-GA_GAA-3) and M2 (5-UUCUUU-3), respectively [56]. These effector protein-RNA tandem complexes decrease the levels of the alternatively spliced versions of *TaXa21-H* and *TaWRKY53*, both encoding proteins that are positively involved in plant adaption to various stresses, thereby reducing the plant defense responses [56].

Besides DNA and RNA, fungal effectors target plant proteins as well, such as the conserved necrosis-inducing secreted protein 1 (NIS1) effector from the fungal endophyte *Colletotrichum tofieldiae* and the rice blast fungus *M. oryzae* [32]. NIS1 interacts with the PRR-associated serine/threonine protein kinases BAK1/SERK3 and the Botrytis-induced kinase 1, inhibiting their kinase activities and preventing the PAMP-triggered reactive oxygen species (ROS) burst, resulting from the PTI activation in *Arabidopsis*, rice, and *Hordeum vulgare* (barley) [32].

## 3. Toward the Establishment of a Functional AM Symbiosis by Suppression of the Host Immunity

The broad conservation of plant defense responses to beneficial and pathogenic microbes, together with their common effectiveness in colonizing host tissues, led to the investigation of genes encoding potential effector proteins in AMF [57]. Plants deprived of inorganic phosphate secrete specific root exudates that trigger the germination of surrounding AMF spores [58,59]. After a tightly coordinated exchange of molecular signals, plants undergo a complex cellular reprogramming that allows AMF to grow from the appressorium through the different plant cell layers that form the intraradical mycelium (IRM), which further penetrates the root cortical cells to give rise to highly branched structures, called arbuscules [60]. These arbuscules are intimate operational interfaces that undergo continuous turnover and are the primary site of nutrient exchange [3,61,62]. Outside the plant root, the fungal extraradical mycelium (ERM) extensively branches to reach nutrient reservoirs that are otherwise inaccessible to the plant roots [63].

Similarly to pathogenic fungi, AMF needs to overcome the plant defense mechanisms to establish themselves inside the roots of the host plants. The intensity of the plant immune response to AMF colonization has been described as transient and weak [64]. AMF hyphal expansion is restricted to some areas of the roots, implying the existence of a plant defense system that controls fungal overgrowth [65]. The perception of chitin-derived fungal molecules by the plant triggers a low oxidative burst with an accumulation of hydrogen peroxide (H_2_O_2_), an important ROS molecule in plant immunity [66], and an increase in antioxidant activity [67]. This process is accompanied by a transient increase in salicylic acid during fungal appressorium formation in *Nicotiana tabacum* (tobacco) roots, hinting at PTI activation at early AM symbiosis phases [67]. Accumulation of H_2_O_2_ has also been reported in hyphal tips and arbusculated cells of *M. truncatula* [68]. In addition, nitric oxide (NO), another signaling molecule implicated in plant defense responses [69], accumulates in the root hairs and epidermal cells of *Solanum lycopersicum* (tomato) roots during the initial steps of the AMF association [70]. The accumulation of NO resulted in a prompt but steady expression of the *phytoglobin 1* (*PHYTOGB1*) gene, involved in NO metabolism, that significantly differed from that of infection with the pathogenic fungi *Fusarium oxysporum*, *Phytophthora parasitica*, and *Botrytis cinerea*, indicating that AMF induces a specific NO signature [70]. One of the reasons for this weak initial plant immunity response may be the low plant cell wall hydrolytic capacity of AMF, which could deploy the degrading enzymes only in a limited area for the appressorium formation [71]. Moreover, a gradual dosage of chitin elicitors from the AMF, as well as the AMF-mediated degradation or scavenging of chitin-derived molecules, could help to mitigate the plant immune responses [30]. This fine-tuned defense reaction may be crucial for the progression and control of the AM symbiosis at different stages of the interaction [72].

Besides immunity, many transcriptional and cellular mechanisms are differentially modulated during the establishment of an AM. For instance, the perception of Myc factors by plant cell membrane receptors triggers the symbiosis pathway, which is typified by the fluctuation of nuclear calcium (Ca^2+^) levels. In *M. truncatula*, this ion spiking is decoded by the Ca^2+^ and calmodulin-dependent protein kinase doesn’t make infection 3 (MtDMI3) that activates an intricate network of TFs to promote downstream events necessary for fungal colonization [73,74,75]. Some of these events, strictly coordinated by the plant nucleus, drive the intercellular invasion via the development of a prepenetration apparatus that consists of a cytoplasmic front enriched in microtubule and microfilament structures, endoplasmic reticulum (ER) cisternae, and a central membranous thread [76]. Another cellular aspect is differentially modulated during AM symbiosis but is also present in other plant-associated beneficial microbes such as *Rhizobium* spp. [77] and the parasitic root-knot nematodes [78,79,80], is the modification of the cell cycle to support microbial progression [81]. During the establishment of AM in *M. truncatula*, ectopic cell cycle activation in the cortical cells and an increase in ploidy levels in arbusculated and neighboring cells have been reported [81,82,83]. Furthermore, an increase in protein synthesis and transporters may be required to maintain the development and functionality of the arbuscules. Indeed, similar to pathogenic fungi, AMF also relies on SWEET transporters specifically located at the periarbuscular membrane to support arbuscule maintenance, such as the *M. truncatula* MtSWEET1b [84]. Therefore, in addition to immunity, other key cellular aspects, such as cell cycle, ploidy levels, and nutrient homeostasis, must be adjusted in the host plant roots to establish a functional symbiosis. These transcriptional, cellular, and physiological changes of the host plant roots can also be elicited by effector proteins during AM symbiosis as is the case of pathogenic fungi.

## 4. AMF Genomes Encode a Wide Effector Repertoire

Of the 240 described arbuscular mycorrhizal species, less than 10% of the genomes have been sequenced and annotated, allowing the subsequent identification of putative effector proteins [85]. The first published AMF genome was that of the model species *R. irregularis* strain DAOM197198 [19,20,22,86], followed by those of *Gigaspora rosea* [24,87], *G. margarita* [88], *R. clarus* [25], *Diversispora epigaea* [89], *R. diaphanus* and *R. cerebriforme* [24]. More recently, de novo annotation of 15 different AMF genomes has been facilitated by the implementation of nuclear sorting coupled with genome sequencing [90]. The *R. irregularis* DAOM197198 has been assigned as the main model AMF in arbuscular mycorrhizal symbiosis studies [19], not only because of the early access to the genomic and transcriptomic data [18,19], but also because the strain is used as a commercial biofertilizer due to its wide host range [91] and its relatively easy propagation in in vitro cultures in the presence of plant roots or upon application of specific fatty acids, such as myristate [92,93].

Despite these advantages, the difficulty of propagating AMF in the absence of a host, the typical asynchronous growth of the different fungal structures during the interaction, and the large genetic variation of AMF hamper the elucidation and functional characterization of AMF effector proteins. Most of the current efforts to identify putative AMF effectors are based on a combination of transcriptomics and in silico prediction tools. In the last decade, a handful of studies have aimed to unravel the effectome of *R. irregularis*, and its close relatives *R. clarus* and *G. rosea* [18,21,23,26]. Comparative in silico research with the predicted effectomes of *R. irregularis* and *R. clarus* identified a set of 18 common putative AMF effectors that shared more than 90% similarity at the protein level [21]. Of these 18 conserved or core effectors, eight had an NLS, highlighting the possible relevance of AMF effectors for translocation to the plant nucleus [21,94]. In another study, the distantly related embryophytes *M. truncatula*, *Brachypodium distachyon*, and *Lunularia cruciate* were used to determine whether putative effector-encoding genes from *R. irregularis* and *G. rosea* are differentially or commonly expressed during AM symbiosis [23]. Bidirectional genomic mapping between the two available annotated genomic repertoires of *R. irregularis* yielded a list of 872 putative effector proteins, whereas 2633 effector genes were predicted for *G. roseae*, of which 33 effectors of *R. irregularis* were upregulated in the three studied hosts and 53 effectors of *G. rosea* showed consistent upregulation both in *M. truncatula* and *B. distachyon* [23]. Altogether, the conserved effector core shared by both AMF species included 45 genes that, based on expression data, participate at different stages of the symbiosis.

Another study with three host plants, i.e., *M. truncatula*, *Allium schoenoprasum* (chives), and *Nicotiana benthamiana* inoculated with *R. irregularis* strain DAOM19718 demonstrated that a large core set of effector genes is upregulated in all plant hosts when compared to each other. To assess the differential expression of *R. irregularis* effector genes in different symbiotic structures, namely ERM, IRM, and arbuscules, laser microdissection was used on inoculated *M. truncatula* roots [26]. This investigation resulted in a total of 310 putative effector genes expressed in fungal structures, of which 86 exhibited an NLS. Of these genes, 120 were expressed systemically in all fungal organs and were listed as *R. irregularis* core effectors, probably involved in AM progression [26]. Furthermore, 52 candidate effectors were expressed specifically in arbuscule-containing cells, whereas one was expressed in the IRM and 66 were specific for the ERM. This study indicated that *R. irregularis* utilizes specific effectors in different fungal structures. The combination of the host- and cell-specific transcriptomes resulted in an updated list of 338 putative *R. irregularis* effector genes [26].

In summary, AMF genomes code for hundreds of putative effector genes, some of which are used by various AMF species and others that appear to be utilized in a more host-specific and tissue-targeted manner.

## 5. Protein Domains Identified in the *R. irregularis* Effector Repertoire

One of the protein domains shared by many putative AMF effectors is the NLS, although other protein domains have also been identified in the putative effectors of *R. irregularis*. For instance, among the effector proteins with homology to other known motifs or domains, those involved in serine-type endopeptidase, MD-2-related lipid (ML) recognition domains, and chitin-deacetylase were enriched [26]. Serine-type endopeptidases are enzymes involved in the cleavage of peptide bonds, thereby hydrolyzing proteins [95]. Therefore, AMF effectors may target plant proteins for degradation to boost the protein turnover rate or to avoid interception by plant enzymes, such as chitinases. Moreover, ML domain-containing proteins have been proposed to regulate diverse biological functions in host immunity and lipid metabolism by interacting with different lipids in bacteria [96]. Thus, the presence of ML-like effectors may imply that AMF could also exploit similar strategies for its benefit during AM symbiosis [26]. Finally, chitin deacetylases are known to be used by other endophytic fungi, such as *Pestalotiopsis* sp., to scavenge chitin-derived molecules released during the initial invasion of the plant hosts to subvert the chitin-induced immunity [97]. Thus, AMF effectors may rely on comparable mechanisms to evade plant recognition or to reduce the plant immune response.

As mentioned above, recent research in the pathogenic field has confirmed the involvement of effector proteins in manipulating not only plant proteins but also the host DNA and RNA [42,57,98,99]. In the published *R. clarus* effectome, one putative candidate effector protein containing a PWWP domain was identified [21]. PWWP domain-containing proteins are known to recognize plant DNA and histone-methylated lysine in the nucleosomes [100]. Therefore, such an effector could potentially bind DNA in the host plant nucleus to regulate plant gene expression during AM symbiosis. In addition to using effectors with protein-binding activity, AMF could also manipulate the host physiology by modulating its genetic and transcriptional machinery in the plant nucleus.

We hypothesized that also *R. irregularis*, like *R. clarus*, may encode effectors with DNA- or RNA-binding domains, but such effector proteins have not been investigated yet or discussed previously [21,23,26]. To detect *R. irregularis* effector proteins that display DNA and RNA binding, we used the latest *R. irregularis* effectome [26] and examined it with the recently updated version of InterProScan 90.0 (https://www.ebi.ac.uk/interpro/search/sequence/ (accessed on 12 September 2022)) [100]. InterPro combines 13 partner databases into a single online resource, and by using predictive modeling the query is analyzed for predicted domains and sites typically identified in other known proteins, which are often experimentally validated [100]. Following this approach, InterPro categorizes them into protein families and identifies the presence of domains and relevant sites, such as IDRs, signal peptides, transmembrane features, and DNA or RNA binding domains in effector proteins [100]. Out of a total of 338 putative effector proteins identified in the integrated host and stage secretome, 178 did not contain any conserved motif or domain, whereas 160 exhibited predicted functional protein domains [26], among which three candidate effectors were identified that could be involved in interacting with DNA or RNA after translocation into the host cell nucleus (Table 1). Two putative effectors, RirG013260 with an S1/P1 nuclease domain [101] and RirG200050 with a WD40 repeat-containing domain [102] had DNA-binding and DNA-regulatory domains, whereas one effector, RirG267270, displayed an RNA-binding domain PUMILIO HOMOLOG 15-LIKE [103]. Thanks to the InterProScan search with updated protein databases, we discovered homology with the RING-H2 zinc finger C3HC4 domain, known to interact with DNA [104], in the jgi.p|Gloin1|346360 effectors (Table 1).

Interestingly, when combining the effector protein homology dataset [26] with the updated InterProScan results, we also identified 12 effectors with predicted functional domains possibly relevant to AM symbiosis, some of which had not been discussed previously (Table 1). Five effectors had domains similar to glyoxal oxidases (jgi.p|Gloin1|161262, RirG257590, RirG043650, RirG180400, and RirG187640) [26]. Glyoxal oxidases catalyze the production of H_2_O_2_, which is required for the correct functioning of lignin-degrading enzymes and could be implicated in the modulation of the plant immune response by fungi [105]. In addition, four effector proteins displayed Ca^2+^-binding domains (jgi.p|Gloin1|154898, RirG043250, RirG045350 and RirG101100). Nuclear Ca^2+^ spiking is an important hallmark of the symbiosis pathway signal transduction, and it is necessary to initiate the AM symbiosis [75,76,106]. Thus, a symbiont effector with such a protein domain could adjust intracellular Ca^2+^ levels in the early stages of the interaction. Three effectors (RirG175680, RirG263220, RirG165580) exhibited different functional domains with interesting features possibly linked with AMF colonization activities. One effector had an expansin-like domain (RirG175680), a protein with such a domain that has been reported to promote fungal accommodation by increasing cell wall loosening [107,108]. The second effector (RirG263220) carried a phytocyanin domain/early nodulin (ENOD)-like protein domain. ENOD proteins are key players in nodulation, another endosymbiosis involving nitrogen-fixing rhizobacteria [109,110] and could play a potential role in modulating the plant-fungus interaction. Finally, RirG165580 displays homology to a nitrogen permease regulator 3/negative regulation of the Target Of Rapamycin (TOR). TOR signaling of host nutrient status is likely to be an important determinant of AM symbiosis development [111] and could be differentially regulated by an effector protein with such homology.

Thus, repeated analysis of the potentially large effector reservoirs may reveal new functional domains and open novel avenues for further functional analysis. Tools, such as Alphafold and related artificial intelligence-based methods will certainly shed more light on the effector functions (https://www.alphafold.ebi.ac.uk/). Consequently, like pathogenic effectors, AM effector proteins may orchestrate key processes, such as signaling, nutrient exchange, and plant cell architecture to fine-tune this association.

## 6. Current Insights into AMF Effectors

Although the first *R. irregularis* genome and the subsequently predicted effectome had already been published 10 years ago [18,19], the detailed functional characterization of *R. irregularis* effector proteins are still lagging, probably due to the absence of stable AMF transformation methods [112,113]. Thus, an in-depth investigation of the involvement of effector proteins in the regulation of the AM symbiosis and plant physiology relies on reverse genetic approaches in host plants, on the characterization of the plant molecular targets, and the identification of effectors triggered downstream of signaling events. To date, only five effector proteins from *R. irregularis* have been studied, and only in one host, the model legume *M. truncatula* (Figure 1). The nucleus-localized secreted protein 7 (SP7) has been implicated in the modulation of immunity through the interaction with the pathogenesis-related ethylene response transcription factor 19 (ERF19) (Figure 1a) [27]. The putative strigolactone-induced secreted protein 1 (SIS1) has been shown to be required for arbuscule maintenance [28] and the nucleus-localized crinkler effector 1 (RiCRN1) for symbiosis progression and arbuscule development [29], whereas the secreted LysM-containing effector (RiSLM) interfered with plant chitin-triggered immune responses by binding to fungal chitin oligosaccharides [30], and the nuclear-localized effector 1 (RiNLE1) impaired histone 2b (H2B) monoubiquitination in arbusculated cells by suppressing the expression of defense-related genes to improve AMF colonization (Figure 1b) [31]. Only for two of them, SP7 and RiNLE1, a host-interacting protein has been found (Figure 1) [27,31], whereas for the other three, RiCRN1, SIS1, and RiSLM, their involvement in AM symbiosis has been elucidated by host-induced gene silencing (HIGS) and/or effector overexpression in plant roots [28,29,30]. Compared to the hundreds of putative effector proteins that have been estimated [18,23,26], these data indicate that there is still a large knowledge gap between effector prediction and their positioning in a plant symbiotic context. Elucidation of the interacting host plant macromolecules through which effectors influence downstream molecular pathways may help us to get closer to understanding the biological role of AMF effectors during the different stages of AM symbiosis.

## 7. Approach to Tackle the Unknown

Here, we will provide a detailed overview of the current methods and biochemical approaches that have been used for the identification and functional characterization of AMF effectors. Additionally, we will review some techniques applied in the field of fungal effector research that might fill the gaps in the functional validation of AMF effectors in a relevant biological context.

### 7.1. FIRST STEP: Bioinformatic Assessment of Effector Features

Although much effort has been made to unravel the effectomes of AMF fungi, further verification of effector-like features is needed to ensure their classification as such. Several pipelines and bioinformatics tools are available to recognize candidate effectors from available genomes [50,114,115]. Moreover, for *R. irregularis* and other fungi, protein features in the effector protein sequence are being investigated by a number of in silico prediction tools. Several fungal secretome databases are already available, such as the Fungal Secretome Database (FSD) and FunSecKB for the identification of putative secretory proteins, but no specific integrated platform for the prediction of the AMF secretome has been published [116,117]. One of the main features to generally differentiate fungal effector proteins is the presence of a short N-terminal signal peptide that targets the protein to the conventional ER-to-Golgi secretory pathway [118]. Nevertheless, fungal effector proteins may also exploit alternative routes for their secretion into the extracellular environment and may lack the signal peptide [41,119,120,121]. In general, a signal peptide contains an N-terminal positively charged region followed by a hydrophobic region and a C-terminal peptidase cleavage site that is usually preceded by three small uncharged amino acids [120,122]. Conventional modeling of signal peptide sequences relies on analytical processes, such as the deep neural network algorithm found in the SignalP4.0 software [123] or the Hidden Markov Model (HMM) statistical approach found in the Phobius tool [124]. To our knowledge, no software has been developed to specifically identify motifs that may be involved in the unconventional secretion of fungal effector proteins [121,125]. Nevertheless, the SecretomeP tool, based on a small set of verified signal peptides from independently secreted mammalian and bacterial proteins, has efficiently predicted the unconventional secretion of the VdIsc1 and PsIsc1 effectors from *Verticillium dahliae* and *Phytophthora sojae*, respectively, suggesting that it could also be implemented into AMF research [41,119,126].

Effectors can act both extracellularly in the apoplast and intracellularly in the host cytosol or specific subcellular compartments [17]. Predicting motifs involved in the intracellular translocation of effector proteins is challenging due to the lack of conserved fungal features [121,127]. Nevertheless, the machine learning tool EffectorP can be used to categorize effector queries as apoplastic, cytoplasmic, or noneffector proteins by comparing the amino acid sequence length, molecular mass, net protein charge, cysteine, serine, and tryptophan content with previously reported fungal secretomes [50,125]. Alternatively, EffHunter identifies canonical apoplastic effectors, based on the secretion, localization, size, and cysteine content [128].

A second level of translocation can be exploited by some non-apoplastic and cytoplasmic fungal effectors, which can be further internalized by mimicking host targeting sequences, such as the NLS, mitochondrial, or chloroplastic transit peptides [129]. An accurate machine learning prediction tool for the detection of transit peptides required for subcellular effector translocation is LOCALIZER [129], although other deep learning machines, such as TargetP or DeepLoc, can be used to discriminate between organelle internalization features [130,131].

Fungal effectors, including those of AMF, are often rich in IDRs and rarely share sequence similarity with other annotated proteins, probably due to the strong selective pressure during evolution [127,132]. Proteins with IDRs account for 70% of the eukaryotic signaling proteins and 42% of the R×LR effectors of the pathogenic oomycete *Phytophthora* spp. [133,134,135]. These IDR-enriched proteins are physically flexible because they lack their secondary structure under physiological conditions and rather fold in a stimulus-dependent manner [133,136]. As the same amino acid region can bind multiple protein partners, acting as central players in protein-protein interaction (PPI) networks, IDRs can be hubs for promiscuous interactions [137]. IDRs have been postulated to take part in effector translocation, innate immune evasion, and host protein mimicry [133]. Based on the presence of this feature among the predicted *R. irregularis* effector proteins [21], IDR-containing effectors may have been positively selected to confer advantages to the fungus, for instance, to adapt the involvement of the effector in different pathways in a time- and environment-dependent manner. Valuable online bioinformatics tools for identifying IDR features based on sequence-derived features include InterProScan and DISOPRED, the latter with a focus on the broad prediction of relevant binding sites within the IDRs of eukaryotic proteins [100,138].

Thus, although the exact prediction of specific hallmarks present in effector proteins is complex due to the extensive lack of similarity to well-benchmarked protein domains, several bioinformatics sources have been implemented through machine learning methods to overcome these problems and generally classify putative fungal proteins, including those from AMF.

### 7.2. SECOND STEP: Functional Validation of Effector Proteins

As the prediction of a protein as a putative fungal effector does not prove that it is one, several preliminary experimental steps must be taken to fully validate the function of the putative AMF effectors during the symbiotic association with their plant host. First, because *R. irregularis* effector genes are differentially expressed in a host- and tissue-dependent manner, the expression of mycorrhizal effector genes must be confirmed in the specific AMF-host symbiotic framework of interest [23,26]. To date, only one tissue-specific transcriptome dataset of microdissected *R. irregularis* fungal structures in colonized *M. truncatula* roots is available [26]. The generation of more detailed gene expression profiling databases in a large number of mycorrhized plant hosts would not only help the AMF effector research but also improve our understanding of AMF host specificity [139,140,141]. Moreover, novel techniques, such as single-cell RNA sequencing, could provide in-depth transcriptome profiling of effector genes in individual colonized cells, whereas spatial transcriptomics could contribute to simultaneously quantify and localize AMF effector expression in a cell- and tissue-specific context [140,142]. However, to our knowledge, although these techniques have been applied for nodulation [143], they have not been implemented in the AM symbiosis framework, probably due to the complex nature of the plant-fungal interfaces created during the interaction and the difficulty to enrich for colonized structures in a non-destructive way.

Second, the in silico predicted domains, such as the signal peptide, NLS, or other known protein domains, should be supported by functional validations. Due to the inability to genetically modify *R. irregularis*, alternative approaches, such as the Yeast Secretion Trap, have been widely used to confirm the conventional secretion nature of AMF effectors. This method relies on the expression of the predicted signal peptide of the putative effector fused to the *SUCROSE INVERTASE 2* (*SUC2*) gene lacking its endogenous signal peptide in a sucrose-deficient *Saccharomyces cerevisiae* strain. A functional signal peptide will secrete the SUC2 fusion protein, allowing the transformant cells to grow by metabolizing the sucrose in the medium [144,145]. As this method is restricted to *S. cerevisiae*, it provides no information on the biological context in which effectors are secreted [145]. Therefore, alternative experimental approaches that use genetically modified fungal microorganisms with similar colonization strategies to AMF can be implemented. Some of these microorganisms include the hemibiotrophic fungi *Phytophthora palmivora* and *M. oryzae*, both form haustoria at the root interface [146,147] and display structural similarities to AMF arbuscules. Indeed, *M. oryzae* has been used to demonstrate the secretion and translocation of the *R. irregularis* SP7 effector protein [27].

To validate their subcellular localization *in planta*, effectors lacking their signal peptide can be fused to fluorescent proteins and transiently expressed in *N. benthamiana* leaves, in *Arabidopsis* protoplasts, or plant hosts. The subcellular localization of three of the five characterized *R. irregularis* effectors has been demonstrated by means of a fluorescently tagged effector version (Table 2). For example, to investigate the *planta* nuclear localization of the RiCRN1 effector protein, fluorescent protein fusions at the C-terminal site of the signal peptide-lacking effector, at the functional region containing the three predicted NLSs and the full-length effector fusion protein were used to validate the nuclear localization [29]. When NLS sites are predicted in an effector sequence, the specific contribution of the NLSs can be examined by deletion or missense mutation analysis [33,148]. Moreover, direct *in planta* identifications by technologically advanced proteomic-based approaches, such as fluorescence-assisted cell and nuclei sorting or laser capture microdissection (LCM) combined with mass spectrometry, to validate the effector localization offers promising prospects [149].

Third, to explore whether fungal effector proteins influence plant development and AM symbiosis, transgenic host plants can be produced in which the effectors are heterologously expressed. The generation of stable transgenic plant host lines is tedious and time-consuming, hence, alternative strategies, such as the production of composite plants, have been preferred [150,151]. Composite plants are usually produced by *Agrobacterium rhizogenes* (*A. rhizogenes*)-mediated hairy root transformation and exhibit a wild-type shoot and a transgenic root system [152]. However, phenotyping of AM fungal colonization in composite plants remains challenging because expression levels can differ among independently transformed roots [31].

In addition to ectopic expression, effector genes can be silenced by HIGS or interference RNA (RNAi). Subsequently, the fungal structures can be quantified and morphologically studied to characterize the involvement of these effectors in the modulation of relevant AM symbiosis traits. This approach has been implemented in the investigation of the five characterized effector proteins in colonized roots of *M. truncatula* listed in Table 2 [27,28,29,30,31]. HIGS and RNAi are based on posttranscriptional gene silencing triggered by double-stranded RNA molecules complementary to the target sequence, which are processed by Dicer-like proteins [153,154]. These siRNAs are then transferred into the microbe, where together with Argonaute, they form the effector RNA-induced silencing complexes (RISCs) that target endogenous effector RNAs for degradation or translational inhibition to reduce the number of active effector proteins [28,29,30,155]. Although the molecular process by which the siRNAs are transferred to the microbe is still under debate, these molecules are assumed to be translocated to the microorganism by ingestion or by vesicle trafficking [153,156,157].

Currently, two main methods are used for macroscopic and microscopic quantification of AM symbiotic structures, commonly known as the Trouvelot [158] and the magnified intersection methods [159]. Both approaches rely on the staining of the fungal cell walls, either by commercial ink or by fluorescent staining of the fungal *N*-acetyl-d-glucosamine with the Wheat Germ Agglutinin (WGA) conjugate [61]. Moreover, for the RiCRN1, RiSLM, and RiNLE1 effectors, WGA staining coupled with quantification of different AM traits according to the Trouvelot or the magnified intersection methods has been used, confirming the role of these effector proteins in AM symbiosis (Table 2) [29,30,31]. However, the staining requires fixation procedures that limit the sample viability and do not allow dynamic visualization of fungal structures [158,159]. The Trouvelot method quantifies the macroscopic presence or absence of fungal structures as well as their presence in the whole root or root sections [158]. In contrast, the magnified intersection method is based on the quantification of the different symbiotic structures according to their position on an intersection template [159]. Although both quantification methods are widely used, computational comparison between the projected ink-stained surface area obtained from the Trouvelot method and the intersection determinations still suggests that the Trouvelot method is the most accurate [160]. As the quantification of the relative proportions of fungal structures can be subjective and biased by personal observations, alternative approaches, such as the deep-learning software AMFinder, have been developed [161]. This system allows automatic image quantification of AMF ink-stained structures with computational neural networks, which may help to circumvent the observer-based biases [161].

Another novel method to detect the presence of AMF structures uses the red pigment betalain that is produced exclusively in AM-containing plant tissues, a method called MycoRed [162]. The MycoRed-stained *M. truncatula* transgenic lines contain multigene vectors comprising the three enzymes involved in the betalain biosynthetic pathway, *CYP76AD*, *DODA*, and *cDOPA5GT*, in which the *CYP76AD* expression is driven by the *M. truncatula* AM-specific promoters phosphate transporter 4 (*MtPT4*) or the arbuscule-specific blue copper protein 1 (*MtBCP1*) [162,163,164]. Similarly, symbiotic root areas can easily be detected by the eye.

In addition to AM phenotyping, transcriptional analysis of well-known fungal and plant genes involved in the AM symbiosis is regularly performed to validate the AM phenotypic observations and to evaluate the status of the symbiotic interaction at the molecular level. Common genes that are used to assess the AMF-plant interaction are, among others, the *R. irregularis* elongation factor 1α (*RiEF1α*) and the arbuscule-specific *PT4* gene for the AMF presence and functional symbiosis, respectively [31,164]. Indeed, the important reduction in arbuscule-containing cells in mycorrhized *M. truncatula* composite plants with impaired expression of the effector *RiCRN1* was further supported by the reduced expression of the Mt*PT4* gene [29]. Likewise, the extensive increase in fungal hyphae in *M. truncatula* lines ectopically expressing the *RiNLE1* was transcriptionally confirmed by the increased expression of the fungal gene *RiEF1α* [31]. Yet, other AM-responsive plant genes specifically involved in arbuscule development, such as the half-size ABC transporter stunted arbuscule (*STR)* or the vesicle-associated membrane protein *VAPYRIN*, may also be of interest to support visual quantification of arbuscule formation [165,166].

Besides influencing the symbiosis, heterologous effector expression could also impact general plant growth and development, which may be difficult to assess in composite plants. Therefore, to investigate whether the AMF effector expression has an effect on plant traits, such as shoot and root growth, stable transgenic lines could be rapidly generated, for example in *Arabidopsis*, for which readily available phenotyping approaches have been published and are routinely used [167,168]. Likewise, if the effector affects a conserved developmental process, the consequences can also be observed in *Arabidopsis*, albeit it is not an AMF host [169,170]. Quantification of plant phenotypic changes can further help elucidate the underlying role of effector proteins during AM symbiosis [27]. Thus far, only an effect on rice growth has been scored for the SP7 AMF effector protein in plants inoculated with *M. oryzae* expressing the SP7 fusion proteins. A decrease in the root decay symptoms typically caused by *M. oryzae* was observed, hinting at a role for SP7 in mitigating plant immune defense responses [27].

### 7.3. THIRD STEP: Seeking the Hidden Interacting Plant Partners

One way that effectors modulate plant physiology is by binding to plant macromolecules and modulating their function. Of the five AMF effector proteins studied, the plant protein targets of only two have been identified [27,31]. Two different approaches were carried out: a yeast two-hybrid (Y2H) screening of SP7 against a mycorrhized root cDNA library of *M. truncatula* and an immunoprecipitation (IP) coupled to liquid chromatography–tandem mass spectrometry (LC-MS/MS) in *M. truncatula* roots ectopically expressing the RiNLE1 fused to a FLAG tag [27,31]. Here, *RiNLE1* expression was driven by the *PT4* promoter to exclusively enrich for arbuscule-specific proteins [31].

Y2H screening is a powerful high-throughput method that allows the identification of direct strong binary PPIs (Figure 2) [171]. Improved Y2H methods include Y2H screening coupled to next-generation sequencing (NGS) (Y2H-seq) and multiplexed Cre reporter-mediated Y2H NGS (CrY2H-seq). In Y2H-seq screening, the abundances of interacting preys can be compared to a background list of false-positive preys [172], whereas CrY2H-seq allows the massive amplification of Cre-recombined positive clones that can be pooled in a single sample, yielding the simultaneous identification of multiple interacting clones [173]. Y2H screening techniques also have limitations, e.g., low-affinity and transient interactions are not easily detected, false positive interactions may result from the high expression levels of the effector bait, and the targeted nuclear expression in a heterologous system may not represent the correct subcellular status of the bait protein [171,174]. As an alternative approach to detect novel interacting plant proteins, IP combined with LC-MS/MS has been used successfully to unravel the RiNLE1 protein target (Figure 2) [31]. IP LC-MS/MS is a sensitive method that allows the identification of effector-plant interacting protein complexes in a physiologically relevant environment when compared to appropriate controls [175]. However, the constitutive expression of effectors in composite plants could affect the sensitivity due to different gene expression levels in different samples. In addition, weak and transient interactions may not be detected due to interference by the protein extraction process, and the number of both false positives and false negatives may increase because proteins are captured after cell lysis [176].

To overcome these drawbacks, alternative methods, such as those using proximity-labeling via effector fusions to a biotin ligase, are valuable tools for de novo discovery of plant protein targets in the bait proximity, the proxeome, and can be used to identify the effector-plant protein network (Figure 2) [177]. The effector protein of interest is fused to an engineered biotin ligase tag, such as BirA or the more sensitive TurboID or UltraID, that biotinylates proteins as close as 10 nm in a timeframe of 10 min up to 24 h [178,179,180]. As biotinylation occurs only in living cells, irrelevant interactions resulting from extraction procedures are avoided, reducing the number of false-positive prey [176]. However, specific controls must be included because spontaneous biotinylation of nearby proteins may still take place, resulting in a list of non-specific and background protein candidates [179,181]. This proximity-labeling approach has already been successfully applied to the identification of translocated effector proteins from the fungal pathogen *Fusarium graminearum* [182] and may therefore represent a relevant alternative for the identification of AMF plant protein preys.

Once the putative interacting plant protein candidates have been identified, additional direct PPI verification assays must be performed. Besides the well-established Y2H pairwise assay that was also used to confirm the interaction of RiNLE1 and SP7 with their plant targets [27,31], many other techniques are suitable to verify the association between the effector and the target plant host protein. For instance, a bimolecular fluorescence complementation (BiFC) assay was performed in *N. benthamiana* leaf cells to determine the nuclear association between SP7 and its interacting partner ERF19 as well [27]. In this assay, effector and plant protein target candidates are fused to two complementary fragments of a fluorescent protein that reconstitutes exclusively upon functional PPI, forming an irreversible fluorophore at the specific subcellular association site [27,183]. An improved version of BiFC is ratiometric BiFC (rBiFC), based on the 2in1 cloning system in which the T-DNA insertion contains the two potentially interacting proteins flanked by the corresponding split fluorophores and an additional epitope tag, as well as an independent RFP cassette as transformation control [184]. Expression of the three fluorescent proteins is driven independently by their constitutive promoters, ensuring equal gene dosage within individually transformed cells. Because of the consistent dosage, the technology facilitates quantification of the fluorescent intensity of the interaction and subsequent normalization to the constitutively expressed RFP, resulting in a ratio that can be compared to positive and negative PPI controls [184]. The presence of individual tags on each fusion protein further enables the co-immunoprecipitation (co-IP) of the protein complexes [184]. Nevertheless, the PPIs assessed through BiFC and its variants are irreversible, and the formation of the fluorescent complex is slow, hampering the monitoring of transient associations and the implementation in dynamic studies [185].

To overcome this limitation, techniques, such as the firefly luciferase complementation could be applied [186]. Similar to BiFC, the firefly luciferase enzyme is cleaved into its N- and C-terminal protein halves and fused to the interacting protein candidates. If association occurs, the enzyme is reassembled and, after the addition of the specific substrate, the luciferase activity can be quantified with a luminometer [187]. Alternatively, the split Fluorescence-Activating and absorption-Shifting Tag (FAST) technology allows live-cell monitoring of the formation and dissociation of split FAST protein fusions, which after complementation, can reversibly bind to the fluorogenic substrate hydroxybenzylidene rhodamine, allowing the visualization of the associated proteins [185].

Effectors might also bind DNA or RNA, in addition to proteins, and affect plant host gene transcription or mRNA translation [42,55] (Figure 2). In *R. irregularis* and *R. clarus* four AMF effectors were predicted to bind DNA or have DNA-binding regions. To elucidate which DNA segments the AMF effectors might target in host cell nuclei, chromatin IP (ChIP) coupled to PCR amplification (ChIP-PCR) or NGS (ChIP-seq) is a valuable tool when potential DNA regions have been predicted or when the binding sites are unknown, respectively (Figure 2) [55,188,189]. Similar to other techniques, ChIP relies on the generation of transgenic host plants expressing an effector fusion protein containing a generic epitope tag, such as FLAG or GFP, that can be subjected to pull-down [188]. DNA-protein complexes are cross-linked, fragmented, and treated with exonuclease to remove unspecifically unbound oligonucleotides. The resulting DNA-protein complexes are then precipitated with specific antibodies, allowing the identification of the DNA targets by PCR or NGS. Further validation approaches are necessary to confirm the formation of the effector protein plant host’s DNA complex, such as yeast one-hybrid (Y1H), electrophoresis mobility shift assay (EMSA), and luciferase reporter assays [42,55,190]. Only one *R. irregularis* effector protein was predicted to bind RNA and, thus, could modulate transcriptional processes, such as alternative splicing [191]. Two straightforward approaches to identifying physical associations between fungal effectors and host RNA molecules are RNA-IP (RIP) and Cross-Linking RNA IP (iCLIP) that can be coupled to NGS (iCLIP-NGS) (Figure 2) [192,193].

Similar to ChIP, RIP is based on the use of a specific antibody against the effector-tagged protein fusion of interest that can be pulled down together with the targeted host RNA complexes [192]. Cross-linking steps can be avoided, allowing the quantification of ribonucleoprotein complexes under physiological conditions in native RIP. However, as with other IP approaches, false-positive interacting RNA molecules may appear after cell lysis, especially those that are overrepresented, such as rRNA transcripts [192]. To circumvent these off-targets and to identify specific nucleotide binding sites, a cross-linking step is included [193], using cross-linking agents, such as UV irradiation for irreversible RNA-protein complex studies or formaldehyde-dependent reversible cross-linking that preserves specific native conformations [193].

Therefore, several PPI techniques have proven to be relevant for the identification and validation of plant targets of different natures of fungal effectors and could be further implemented in AMF research to elucidate the molecular pathways by which effectors affect plant physiology.

## 8. Overview and Future Prospectives

Effector proteins are important players in beneficial and pathogenic plant-microbe interactions, where they aim to manipulate plant host defense and physiology by associating with plant host DNA, RNA, and proteins. The recent effectome of the AMF *R. irregularis* encodes approximately 338 putative secreted effectors expressed during symbiosis, the majority of which contain features related to protein binding, although some may bind other host biomolecules, such as DNA or RNA. Thus far, only five *R. irregularis* effectors have been investigated in detail, and for just two, a host protein partner has been identified, highlighting the large gap between the effector in silico prediction and the subsequent functional characterization during the plant-microbe interaction. The lack of information on the role of AM fungal effectors during symbiosis is mainly challenged by three factors: (i) the inability to genetically modify *R. irregularis*, (ii) the difficulty to obtain detailed expression data from the different stages of AM symbiosis in different hosts, and (iii) the lack of sequence similarity of AMF effectors with known proteins that hinders straightforward in-depth functional characterization. Therefore, many problems need to be overcome to unravel the biological role of AMF effectors during the interaction with diverse plant hosts. Here, we review not only the previous techniques that were used in AMF effector research to date but also other promising unambiguous approaches currently carried out in the fungal effectome field that may be useful. The generation of more detailed transcriptomic studies complemented by novel proteomics-based methodologies, jointly with the implementation of reverse genetics and detailed phenotyping strategies, will allow the identification in more physiologically relevant contexts and will help to elucidate the underlying function of AMF effectors during symbiosis and plant growth. Such knowledge might open up new venues for the development of agricultural products aimed to increase symbiotic plant fitness and yield, helping to cope with the food insecurity that we are facing due to global warming.

## Figures and Tables

**Figure 1 ijms-24-09125-f001:**
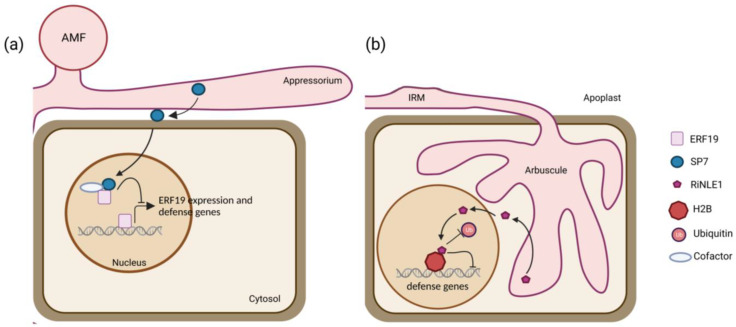
Mode-of-action of the characterized nuclear-localized effector protein SP7 and RiNLE1. (**a**) The *R. irregularis* effector protein SP7 is secreted from the fungal hyphae, translocated to the plant cell, and internalized into the host nucleus. Then, it binds to the ERF19 to suppress the expression of pathogenesis-related genes in *M. truncatula*. (**b**) RiNLE1 is secreted from the arbuscule and further compartmentalized to the plant nucleus where it binds the *M. truncatula* H2B, hindering its ubiquitination to reduce the expression of plant defense-related genes.

**Figure 2 ijms-24-09125-f002:**
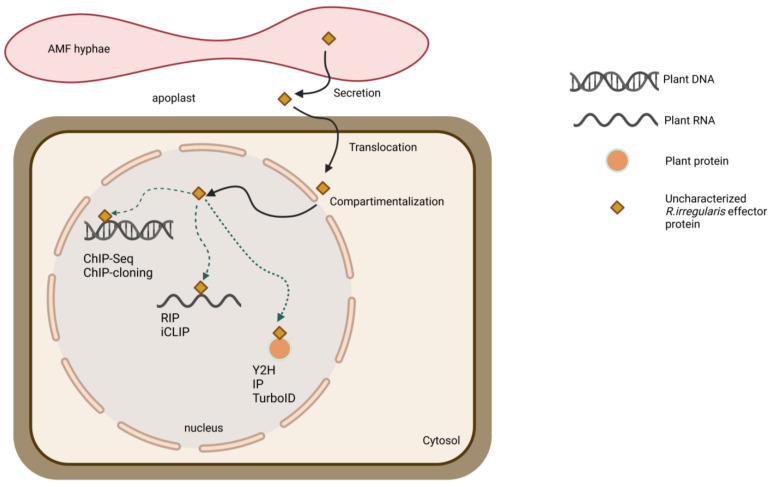
Representation of the main nucleus-localized effector-related processes and the principal wet lab techniques used for the identification of plant targets. AMF effectors are synthesized and processed for signal peptide-directed secretion to the apoplast where some of them translocate to the intracellular space and further to other compartments, such as the nucleus. Once in the plant cell, the effectors associate with RNA, DNA, or RNA to modulate their activity. Although PPI techniques, such as Y2H and IP have been proven useful for the identification of plant proteins targeted by AMF effectors, alternative methods that include protein proximity labeling can be exploited. No AMF effector with DNA or RNA manipulating function has been studied yet. Nevertheless, DNA-targeted sequences by other fungal effector proteins have been identified with ChIP and its derived variants, whereas RNA-bound regions can be studied by means of RIP and iCLIP.

**Table 1 ijms-24-09125-t001:** Putative secreted effector proteins of *R*. *irregularis* and their in silico predicted protein domains.

*R. irregularis* Effector ID ^a^	InterProScan Protein Domains
RirG175680	RlpA-like protein, double-psi beta-barrel domain/expansin
RirG165580	Nitrogen permease regulator 3/negative regulation of Target Of Rapamycin signaling
RirG263220	Phytocyanin domain/Early nodulin-like protein domain
RirG200050	WD40 repeat-containing domain superfamily/Armadillo-like helical protein
jgi.p|Gloin1|346360	RING-H2 Zinc finger C_3_HC_4_
RirG013260	S1/P1 nuclease domain superfamily
RirG267270	RNA-binding protein/Plant self-incompatibility S1/Pumilio homolog 15-like
jgi.p|Gloin1|154898	Calcium/lipid-binding domain/tricalbin
RirG043250	EF-hand domain/Calcium-binding protein
RirG045350	Calreticulin/calnexin calcium-binding ER chaperones
RirG101100	Calcium-dependent phosphotriesterase
RirG043650, RirG257590, RirG187640, RirG180400, jgi.p|Gloin1|161262	Glyoxal oxidase

^a^ Previously reported effector protein domains from Zeng et al. [26] are noted in green, whilst effectors displaying homology with updated InterProScan protein databases are highlighted in blue.

**Table 2 ijms-24-09125-t002:** Summary of the approaches used to functionally characterize effector proteins from *R. irregularis* in *M. truncatula* roots.

EP	Secretion		Subcellular Localization		Plant Target		Ref.
	Method	Results	Method	Results	Method	Results	
SP7	YST and *M. oryzae* SP7-mediated secretion	Secreted and translocated to the cell nucleus	Transient expression; *Agrobacterium*-mediated infiltration of *N. benthamiana* leaves	Nuclear	Y2H screening	ERF19	[27]
SIS1	Not tested	Not tested	Not tested	Not tested	Not tested	Not tested	[28]
RiCRN1	Not tested	Not tested	Transient expression; *Agrobacterium*-mediated infiltration of *N. benthamiana* leaves	Nuclear bodies	Not tested	Not tested	[29]
RiLSM	YST	Secreted	Not tested	Not tested	Microscale thermophoresis	*R. irregularis* CO 4, 5, and 7	[30]
RiNLE1	YST	Secreted	Ectopic expression in *M. truncatula* composite plants	Nucleolar and nuclear bodies	IP-LC MS/MS on *M. truncatula* arbusculated cells	MtH2B	[31]

## Data Availability

Not applicable.

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
