# Peer review of "Characterization of Arbuscular Mycorrhizal Effector Proteins"

_ijms, 2023, doi:10.3390/ijms24119125_

Round 1

Reviewer 1 Report

*In line 136-139 at section 2 of the manuscript, the authors have written IN SUMMARY,... In this part of the manuscript, authors do not need to give any conclusion nor any suggestions. This paragraph should be re-written.

*In section three, how many times authors give the explanation of AMF that they use Reference 1 in this section. This sentence (line 144-146) should be deleted as it has been mentioned already in Introduction part.

*Authors do not need to use BOLD typing for Tables and Figures in the manuscript. For example, check Table 1 in line 297.

*At Table 1, R. irregularis should be Italic.

*Please, correct the paragraphing after Table 1, line 495.

*Line 509, before References 27-31, Table should be written in the manuscript, not in the parentheses.

*Lines 576-578, (7.2), it is recommended authors give their conclusion and their findings contributed to the results and findings of other researchers in other specific section. Maybe, they can design the manuscript with DISCUSSION part or/and give clear conclusion and suggestions in Conclusion part.

*This article also needs one Abbreviation part before References, and authors should mention all Abbreviation of this manuscript which are many in this section for clear understanding of the readers.

*The DOI of all articles in References should be added in the manuscript. Also, all references and citations should double-checked with the references in the manuscript. 

The manuscript need Minor English revision. Some sentences and paragraphs are not clear, and some parts of the manuscript has been repeated, which should be revised carefully.

Reviewer 2 Report

I would like to suggest minor updates.

In the title, the extra words " Guideline to the" should remove. 

The functional mechanism is not present for various traits. the flow chart or step-by-step activity regarding characterization. 

commercial prospective may include.

the minor spelling mistakes may correct.

Reviewer 3 Report

The review presented by Chacon and the authors reviews research on the molecular mechanisms of fungal colonization of plant roots, with an emphasis on proteins. In addition, in silico and experimental methods for functional characterization of effector proteins are presented. This is a very interesting area for research, although it is still insufficiently explored. The paper is well written in a readable and clear manner, the topic is thoroughly covered, and relevant and recent references are given. There are only a few minor points that the authors should consider:

·         Line 295-303. it would be useful if the authors elaborate and discuss the criteria for selecting these proteins as putative host protein/DNA interaction partners. What is the percentage of homology with the corresponding host domains or other domains known to make these interactions?

·         Table 1: Species name and in silico should be in italic

·          It is quite curious that most (but not all) full protein names are written in capital letters. Perhaps it is also not necessary to write the full names, as abbreviations are standardly used and it is confusing when reading the text?

·         At the end of the manuscript is an additional table, which was probably left there by mistake?
